# Exploring the Role of Metabolic Hyperferritinaemia (MHF) in Steatotic Liver Disease (SLD) and Hepatocellular Carcinoma (HCC)

**DOI:** 10.3390/cancers17050842

**Published:** 2025-02-28

**Authors:** Nikolaos-Andreas Anastasopoulos, Alexandra Barbouti, Anna C. Goussia, Dimitrios K. Christodoulou, Georgios K. Glantzounis

**Affiliations:** 1HPB Unit, Department of Surgery, University Hospital of Ioannina, 45110 Ioannina, Greece; 2Imperial College Renal and Transplant Centre, Imperial College Healthcare NHS Trust, London W12 0HS, UK; 3Department of Anatomy-Histology-Embryology, Faculty of Medicine, School of Health Sciences, University of Ioannina, 45110 Ioannina, Greece; 4Department of Pathology, University Hospital of Ioannina, 45110 Ioannina, Greece; 5Department of Gastroenterology, University Hospital of Ioannina, 45110 Ioannina, Greece

**Keywords:** steatotic liver disease, metabolic-associated steatotic liver disease, metabolic-associated steatohepatitis, hepatocellular carcinoma, iron metabolism, dysmetabolic iron overload syndrome, oxidative stress

## Abstract

Iron overload and aberrant distribution in the hepatic parenchyma drive the development and progression of steatotic liver disease to hepatocellular carcinoma. Different molecular mechanisms involving oxidative stress and ferroptosis become activated during this spectrum of disease, starting from lipid accumulation in the hepatocytes (simple hepatocellular steatosis), to reinforcement of this phenotype from micro- to macro-vesicular, to a pro-inflammatory milieu with activated Kupffer and stellate cells, fibrosis, and ultimately hepatocarcinogenesis. Understanding the points where iron plays a key role in this process can help harness its therapeutic potential.

## 1. Introduction

Recent global data demonstrate that primary liver cancer represents the sixth most common type of malignancy, while it accounts for approximately 750,000 deaths annually, rendering it the third most common cause of malignancy-related mortality in the year 2022 [1,2]. Hepatocellular carcinoma (HCC) accounts for almost 80% of the cases of primary liver cancer globally, with significant local epidemiology variations [3]. Both incidence and mortality rates for HCC have climbed despite recent advances in diagnosis, management, and therapeutics in the field. Changes in the global epidemiology of hepatitis B and C, due to vaccination programmes and the success of direct-acting antivirals (DAA), combined with an ageing population with an increasing prevalence of Metabolic Syndrome (MS), have shaped a new scenery in the epidemiology of liver cancer [4].

Interestingly, the uptrend in MS and the resultant rise in HCC-related cases has led to a focused interest in the intricate relationship between fatty liver disease and HCC, with a US incidence of HCC in SLD of 0.72 per 1000 person-years [5], and Asian cohorts estimating that between 0.18 and 0.64 person-years [6,7]. The term has evolved from Non-Alcoholic Fatty Liver Disease (NAFLD) [8], which reflected the low alcohol consumption and lack of other comorbidities in this population, to Metabolic-Associated Fatty Liver Disease (MAFLD) [9], recognising the predominant patient pattern of MS, to the modern umbrella term, Steatotic Liver Disease (SLD) [10], which includes Metabolic Dysfunction-Associated Steatotic Liver Disease (MASLD), MASLD with increased alcohol intake (MetALD), Alcoholic Liver Disease (ALD), Cryptogenic SLD, and other causes. The novel terminology stemmed from a deeper understanding of the pathophysiology of this disease, namely as a self-propagating cycle of cause and effect between MS and lipid accumulation in hepatocytes, mediated and facilitated by aberrant cellular metabolism [11]. Inevitably, the changes in hepatocytes lead to the activation of multiple cellular pathways, which impact not only the hepatic microenvironment but also the malignant potential of the hepatocyte itself, with chronicity leading to the genesis of HCC. These mechanisms include, but are not limited to, oxidative stress, cellular senescence, autophagy, and ferroptosis [12,13,14].

There are multiple sources in the literature linking iron with this pathological cascade and which have attempted to explore its role in this sequela [15,16]. Iron is a key element for almost all living cells and organisms, as it engages in multiple critical biochemical processes and participates in several enzymatic reactions. However, when present in excess within cells and tissues, iron serves as the main catalyst for the generation of reactive free radicals, which cause uncontrolled oxidations to essential biomolecules, leading to oxidative stress [17]. Therefore, mammals are equipped with a constellation of regulatory mechanisms, so they can fulfil their metabolic needs for iron and at the same time minimise its toxic effects [18,19]. Previous studies have associated hyperferritinaemia, dysfunctional iron metabolism, and storage with fatty liver disease generation and progression as well as with HCC-genesis [20,21]. In this review, we aim to summarise the available evidence and provide updates on the contribution of metabolic hyperferritinaemia to the pathophysiology of hepatic steatosis and HCC.

## 2. Iron Metabolism and Metabolic Hyperferritinaemia

### 2.1. Physiological Iron Metabolism

Iron has a well-recognised role in the maintenance of multiple normal homeostatic mechanisms and biological processes, including DNA synthesis via ribonucleotide synthase, cellular respiration, enzymatic reactions, and oxygen transport and tissue delivery (haemoglobin and myoglobin), and is available as ferrous (Fe^2+^) or ferric (Fe^3+^) in humans. Excess iron can lead to cellular iron overload, help generate reactive oxygen species (ROS) via the Fenton reaction and can lead to tissue iron deposition and end-organ damage with multiple mechanisms. Thus, a constellation of complex mechanisms is employed to tightly regulate total and cellular iron levels [12,17].

Dietary ferric iron is reduced to ferrous and then absorbed in the duodenum by the apical membrane Divalent Metal Transporter 1 (DMT1), while its release to the bloodstream is facilitated by ferroportin. When ferrous iron is due to be exported, it is re-oxidised (in the cytoplasm by ceruloplasmin, or on the cellular membrane by hephaestin) to allow for binding to transferrin and subsequently its transport for utilisation. For most cells, ferric iron is introduced via endocytosis, triggered by binding of the transferrin to its type 1 receptor (TfR1), and subsequently ferric iron is reduced and released to the cytosol for multipurpose use [17,19].

A key player in the regulation of iron distribution is ferritin, an intracellular, spherical iron storage protein consisting of varying proportions of 24 heavy- and light-chain molecules, with an iron storing capacity of up to 4500 iron ions as ferric oxy-hydroxide phosphate. The key difference between the heavy and light chains is their speed and ability to release iron, with the heavy chain being able to mobilise and store faster than the light chain, the latter being more useful in long-term iron storage in hepatic and splenic cells. Of note, the heavy chain also possesses ferroxidase activity, which is essential for the safe storage of iron in its ferric form, from its ferrous form, readily available in the cell [17,22]. Low cellular iron levels lead to its release from the ferritin molecule via a process known as ferritinophagy, a selective autophagy process mediated by Nuclear Receptor Activator 4 (NCOA4) [23].

Ferritin regulation is part of a larger intracellular iron control system largely reliant on the iron regulatory proteins IRP1 and IRP2, which can bind to iron-responsive elements (IRE) in the untranslated regions (UTRs) of several mRNAs, including those encoding TfR1, ferritin, and ferroportin. Interestingly, both ferritin and ferroportin mRNAs contain a single IRE in their 5′ UTR. In iron-starved cells, when IRPs bind to that IRE, ferritin and ferroportin translation is inhibited, leading to the prevention of iron storage or efflux, increasing the metabolically available intracellular iron. On the other hand, in iron-replete cells, the lack of IRE/IRP interactions allows for ferritin and ferroportin synthesis, and thus excess iron is stored in or exported from the cell. Another player in ferritin regulation is hepcidin, a liver-produced peptide, whose active form allows for ferroportin degradation in conditions of systemic iron overload and subsequently a higher expression of ferritin to store the excess intracellular iron [17,18,19].

### 2.2. Iron Overload—Metabolic Hyperferritinaemia–Labile Iron and Oxidative Stress

The complex mechanisms described above can efficiently control systemic and cellular iron homeostasis. However, for multiple reasons, including genetic diseases, increased dietary intake and absorption, serial blood transfusions, metabolic dysfunction, and others, iron overload can occur [24]. The liver plays a crucial role in iron homeostasis, storing approximately 25–30%, and is affected majorly as a site of preference in iron overload status [25].

The small fraction of unshielded, chelatable intracellular iron with active redox properties is known as a labile iron pool [17]. The multiple ways ferrous iron can be introduced in the labile iron pool include introduction via SLC39A14, a transmembrane iron transporter, heme-oxygenase 1 (HO-1) release of heme-contained iron, ferritinophagy, and more and could showcase different biological processes that are the result of defending mechanism against iron overload [26]. While in normal homeostasis its full spectrum of normal biological functions is poorly characterised, it is known that labile iron has a role in cellular signalling via ROS generation, a process that leads to different types of cellular death pathways when dysregulated [12].

There is a well-known connection between aberrant iron metabolism and insulin resistance with deranged lipid and carbohydrate metabolism, as it has been shown that iron overload can impact both insulin production and resistance due to pancreatic β cell sensitivity [27,28]. The term “Dysmetabolic Iron overload Syndrome” (DIOS) has been previously used to describe pathophysiological co-relation, emphasising the histological evidence of hepatic tissue iron overload [29,30]. More recently, “metabolic hyperferritinaemia” (MHF) has been adopted as an umbrella term, including earlier stages of dysregulated iron metabolism and insulin resistance [31]. It has been shown that not all patients with MASLD and insulin resistance will develop MHF, but genetic predisposition might be the key stakeholder, creating two subsets of SLD patients [31]. Interestingly, serum ferritin levels reflect cellular and total body iron [32], along with significant progress in quantifying liver iron and fat content/degree of steatosis with novel imaging techniques, allowing for a shift in the diagnosis of MHF from biopsy to non-invasive techniques [33,34,35,36]. In a recently published consensus statement from an international panel of experts, three stages of MHF can be identified: stage 1 (metabolic hyperferritinaemia), stage 2 (dysmetabolic iron accumulation), and stage 3 (dysmetabolic iron overload) [31]. This new definition has already been linked with worse hepatic clinical outcomes in the general and MASLD populations [37].

## 3. MHF in Steatotic Liver Disease and HCC

### 3.1. MHF in SLD

#### 3.1.1. Animal Models of MASLD and MASH Shed Light on the Pathophysiological Role of Hepatic Iron Overload

Multiple animal models have been reported in the recent literature exploring the connection between steatosis (secondary to a high-fat diet; a relatively accurate model of human MASLD) and aberrant iron metabolism. Interestingly, there is a common theme of linking hepatocellular lipid accumulation with hepatic iron overload [38,39,40]. The distribution pattern observed involves mainly the reticuloendothelial system (RES) cells (Kupffer cells and sinusoidal macrophages) with hepatocellular iron depletion [41]. Furthermore, in the context of murine MAFLD, Gao et al., showed that hepatocyte iron deficiency, with iron being exported in extracellular vesicles to hepatic stellate cells (HSC), propagates hepatic insulin resistance and lipogenesis via Hypoxia-Induced factor 2α/Activating transcription factor 4 (HIF2α/ATF4) signalling, and the iron-overloaded HSC drive oxidative stress progression to MASH [42]. A high-fat diet has been shown to increase ferritin-heavy and light-chain mRNA transcription while downregulating hepcidin expression [43]. Moreover, increased dietary iron intake seems to aggravate the steatotic impact of a high-fat diet in mouse models [44,45], showcasing suppression of sirtuin 1 (SIRT1) expression, mediated by miR34a in its 3′UTR [44], a sensor of hepatic lipid content. SIRT1 regulates the expression of multiple proteins involved in lipid homeostasis, such as Sterol Regulatory Element-Binding Protein 1 (SREBP1), Nuclear Factor kappa-light-chain-enhancer of activated B cells (NF-κB), Peroxisome proliferator-activated receptor alpha (PPAR-α), gamma co-activator 1 alpha, and more [46].

The interconnection between iron metabolism and hepatic steatosis in a dysmetabolic context is demonstrated by the variety of mechanisms by which several dietary supplements attenuate hepatic steatosis and improve lipid metabolism in animal models. Buriti oil has been shown to increase glutathione peroxidase (Gpx) activity and reduce iron overload [47] and betaine has been shown to reinstate normal hepatocellular iron metabolism [43]. Erchen decoction improves hepatic steatosis by reducing SREBP1 expression (suppressing de novo lipogenesis) [48], reinforcing the proven connection of SREBP1a/c with hepcidin expression, hepatic iron accumulation, and fatty liver disease [49]. Its administration in mice has been shown to increase caveolin 1 expression [48], previously proven to facilitate ferric iron conversion, increase ferritin chain expression, and reduce labile iron as well as regulate iron distribution from heme decomposition [50]. Extra virgin olive oil was shown to reinstate PPAR-a activity and reduce hepatic steatosis in rats fed iron-rich diets via restoring cellular antioxidant potential [51]. Last, cumin seed powder dietary supplement in high-fat diet-fed rats showed a lower degree of hepatic steatosis, inflammatory cell infiltration, liver iron content, and a preserved antioxidant hepatic reservoir [52]. All the abovementioned converge in a single common ground: the regulation of balanced iron distribution amongst different types of cells in the liver and the maintenance of the hepatic antioxidant potential in inhibiting steatotic liver disease development.

Iron chelators seem to demonstrate a consistent effect in attenuating fatty liver disease in multiple animal model studies. Dietary iron overload combined with a high-fat diet (HFD) induced more severe hepatic damage than HFD only, which was alleviated by deferoxamine [53], which has also been shown to have similar anti-inflammatory impacts when administered intraperitoneally [54]. Another study investigated the effect of deferoxamine on the steatotic liver in both mice and cell lines and showed that extracellular iron reduction reversed ferroptosis features and reduced hepatocyte lipid deposition [55]. The combined action of deferoxamine and deferiprone shows a significant reduction in oxidative stress-related ischaemia reperfusion injury in rabbits, hinting at potential benefits in future research for steatotic liver disease reversal [56].

Changes in iron distribution between different types of cells within the liver could drive the progression of MASLD to MASH, as shown in different animal models. Salaye et al. showed that increased dietary iron intake in an HFD mouse model increased the expression of pro-inflammatory cytokines and growth factors via TGF-β activation, while in vitro exposure of HSC to palmitic acid and iron increased the expression of alpha-smooth muscle actin (α-SMA), a marker of activation of myofibroblasts, aggravated by the presence of fatty HepG2 cells [57], while Gao et al. confirmed the same findings in vivo mediated by an excessive ROS mechanism as the culprit to MASH fibrosis [42]. The impact of the amount of dietary iron is highlighted in the complementary results of the studies conducted by the teams of Kitamura et al. and Fujiwara et al.; the former showed attenuation of steatosis and inflammation in an HFD model of iron supplementation [58] whereas the latter, using dietary iron overload, showed a pro-inflammatory hepatic milieu characterised by M1-polarised macrophages, potentially leading to HSC activation [41]. The same macrophage polarisation changes had been previously demonstrated in another high iron diet model of murine MASLD, where the presence of reticuloendothelial system iron and pro-inflammatory IL4 reduced the presence of M2-polarised macrophages via a STAT6 phosphorylation pathway and enhanced M1 polarisation [59]. Both teams of Gao et al. [42] and Handa et al. [59] showed a reversal of fibrosis when the iron chelator deferoxamine was used in their models. Furthermore, increased expression of hepcidin (and hepatocellular iron storage) attenuated the pro-inflammatory cytokine expression and HSC activation of MASH in a rat model [60]. Last, splenocyte iron accumulation maintains MASH via portal vein ROS and Tumour Necrotic Factor alpha (TNF-α) [61]. This study showcases that extrahepatic iron overload might play a role in liver fibrosis and inflammation in SLD, apart from non-hepatocellular hepatic iron.

Recent research activity has focused intensively on the role of ferroptosis in the initiation of MASLD and its progression to MASH, demonstrating missing parts in our understanding of disturbances in iron metabolism in SLD [62]. Ferroptosis, a recently characterised type of cellular death, occurs when iron-mediated lipid peroxidation cannot be offset due to the depletion of glutathione and Gxp4 inhibition. It is characterised by intracellular Fe2+ accumulation and the presence of lipid peroxidation toxic products, such as malondialdehyde (MDA) and 4-hydroxynonenal (4-HNE) [25,62]. As already explored, iron overload and lipid accumulation are key characteristics of SLD and indispensable components of the ferroptosis process. A link between the two, SLD and ferroptosis, seen under the abovementioned lens, becomes obvious when autophagy comes into play, as the excess iron and lipids trigger ferritinophagy and lipophagy, respectively, initiating lipid peroxidation and voiding cellular antioxidant capacity [63]. This leads inadvertently to excess ROS and ferroptosis-driven progression to MASH [64]. Furthermore, suppressing different elements of the ferroptosis mechanism seems to halt progression or even alleviate MASH, such as restoring Gxp4 potential either via HO-1 [65] or bone morphogenetic factor 4 (BMP4) [66] expression, which facilitates iron metabolism and distribution, allowing the cell to recover its antioxidant capacity, or by icariin supplementation that replenishes Gxp4 [67].

#### 3.1.2. Clinical and Observational Studies Confirm the Interconnection of Iron Metabolism and SLD

The risk of developing SLD and progressing through its stages has been documented in large-scale general population studies. Of interest, data from the National Health and Nutrition Examination Survey (NHANES) from 1999 to 2020, with different definitions of SLD applying, construct a biochemical phenotype of patients at risk for NAFLD/MAFLD, who present with lower iron, higher ferritin, and lower transferrin saturation levels, whereas the risk for hepatic fibrosis was higher for the same biochemical profile patients from the USA [68,69,70]. Meanwhile, in a population-based Korean study, MAFLD was again correlated with higher serum ferritin levels [71]. In a central European cohort of the general population aged 40 to 77, 1111 individuals (13% of the cohort) were diagnosed with hyperferritinaemia, while 81% of them were classified as cases of MHF accounting for 10.7% of the cohort [72]; in a large-scale MASLD US cohort, 20% of the patients presented with MHF, with worse cardiovascular and liver-specific survival outcomes [73]. Furthermore, a growing body of hepatic steatosis studies showcase the correlation between aberrant iron metabolism and various components of the MS, altogether constructing a patient profile frequently seen in clinical practice, with a prominent example of a Chinese cohort of Type 2 Diabetic patients, where statistically significantly higher levels of iron and ferritin in the presence of MASLD were linked with higher levels of insulin resistance [74]. Some of these studies are summarised in Table 1, which shows patients’ clinical and biochemical profiles.

Another interesting connection between iron metabolism and MASLD is revealed when studying nutritional habits and obesity. In a Taiwanese study of 208 patients associating dysregulated iron metabolism and dietary patterns, a Western-type diet (rich in processed food and animal-derived fat) was linked to aberrant iron metabolism, central obesity with increased visceral fat, and sarcopenia [75]. Interestingly, animal-derived iron was strongly linked to an increased risk of NAFLD, whereas a high daily intake of plant-based iron had the opposite effect [76]. These findings were confirmed in a larger Chinese cohort but only for male patients [77], accentuated in obese or high-fat-diet-consuming individuals [76]. Further evidence linking central obesity with deranged ferritin and hepcidin in the context of NAFLD is shown by Pan et al. [16], particularly for female patients, as well as by the Fatty Liver in Obesity (FLiO) study, with a model of glucose, serum ferritin, and alanine aminotransferase having an Area Under the Curve of 0.83 in 112 obese and overweight patients [78]. A possible explanation for this link lies with the gut microbiota, as it has been shown that in NAFLD obese patients with raised serum ferritin and increased liver fat were associated with an overall poor diversity and functionality gut microbiome, as well as certain types of colonic bacteria that can influence iron and fat metabolism at the transcriptional level, showing only the tip of the iceberg for this complex nexus of interactions [79].

Apart from the previously described biochemical profile of risk NAFLD patients, which consists of tests commonly performed in everyday practice, several other iron metabolism markers have been linked with SLD. Some cohort studies have identified ceruloplasmin as another surrogate marker for hepatic iron overload in liver steatosis patients, with an inverse trend to ferritin regarding SLD severity and progression. Several studies from different ethnic and genetic backgrounds confirm lower serum ceruloplasmin is associated with increased liver iron deposition, more severe steatosis, and fibrosis, as well as sinister histological features [80,81,82]. Another marker that has been tested is hepcidin, with conflicting results. While increased hepcidin and lower haemojuvelin have been associated with advanced fibrosis in a Turkish cohort of NAFLD patients [83], serum and mRNA levels of hepcidin failed to predict steatohepatitis development of liver fat content in another Swedish cohort [84], with a recently published systematic review and metanalysis of high heterogeneity showcasing higher hepcidin levels in NAFLD [85]. Clarification of the causality of all the abovementioned associations of clinical characteristics with biochemical markers comes from a growing body of recently published genetic studies, all highlighting the bidirectional causal relationship between genetic predisposition to MASLD with higher ferritin levels, and vice versa, the genetic predisposition to iron accumulation with fatty liver disease [73,86,87].

Moreover, as discussed earlier in laboratory models, clinical outcome differences in NAFLD have been shown in multiple studies of diverse populations regarding hepatic iron distribution. In patients with RES iron distribution, higher staining for MDA and a lower proportion of apoptotic cells confirm the role of ferroptosis in the progression to MASH and highlight the role of oxidative stress [88], which are linked with higher rates of hepatic and cardiovascular incidents in these patients, when compared to hepatocellular iron deposition, with patients that have a mixed type of distribution experiencing the highest rates of events, possibly due to cumulative hepatocellular and RES OS activity from iron overload [89]. Similarly, Buzzetti et al. demonstrated that a mixed iron deposition pattern is associated with progression to MASH, and ferritin levels are a marker of fibrosis stage in these patients [90].

Last, the impact of several interventions with varying degrees of invasiveness on the iron metabolism of SLD patients has been studied over the last decade and serves as a surrogate marker for the role of iron in SLD pathophysiology. Caloric restriction with a shift from a red-meat-based to an increased fibre diet led to reduced serum ferritin and liver fat content irrespective of weight loss in a German randomised dietary intervention [91]. While the pathophysiological connections between iron and hepatic steatosis have not yet led to the design of a clinical trial on the impact of iron chelation on SLD, the effect of phlebotomy has been studied multiple times, with initially promising results in reducing hepatic steatosis and insulin resistance in observational studies [92], but not confirmed in a prospective, randomised clinical trial [93]. Serum iron and ferritin were correlated with the presence and severity of steatotic liver disease in a Spanish cohort of obese patients offered laparoscopic sleeve gastrectomy, whereas intraoperative biopsies did not show any statistically significant difference in iron accumulation between steatohepatitis patients versus their counterparts with normal liver architecture [94]. Similar results in pre- and post-operative changes in serum iron markers were also shown in an obese Chinese population; serum iron was correlated with visceral adiposity and hepatic steatosis severity, with statistically significant reduction observed in both serum iron and MAFLD severity when laparoscopic sleeve gastrectomy was offered [95]. All the abovementioned results serve as concrete clinical data to connect iron overload and SLD.

**Table 1 cancers-17-00842-t001:** Clinical characteristics and serum markers of iron, lipid metabolism and liver function tests in human studies of SLD. BMI expressed in kg/m^2^, HbA1c: glycated haemoglobin, ALT: alanine aminotransferase, AST: aspartate aminotransferase, ALP: alkaline phosphatase, TC: total cholesterol, TRG: triglycerides, CP: ceruloplasmin, NAS: NAFLD activity score, NR: not reported. All differences reported from all the below-mentioned studies are statistically significant with a *p*-value lower than 0.05.

Author, Year	Population Size	BMI	Diabetes Mellitus	Liver Function Tests	Iron Metabolism	Lipid Metabolism
El Nakeeb et al., 2017 [96]	113	Not reported but correlated with hyperferritinaemia in both fibrotic and non-fibrotic SLD	HbA1c: non-fibrotic SLD: 5.31 ± 1.40%, fibrotic: 5.24 ± 1.27%, *p* = 0.190	AST and ALP—higher in fibrotic SLD	Serum iron—equal in all groups, serum ferritin: higher in fibrotic SLD vs. healthy	TC and TRG are higher in non-fibrotic SLD vs. healthy
Wang et al., 2022 [81]	138	27.41 (24.67–30.87)	Non-diabetic cohort	Normal ALT 34%, Normal AST 61%	Lower CP ratio linked to lower liver iron	TC: 4.92 (4.19–5.59), TRG: 1.7 (1.22–2.2)
Xia et al., 2023 [82]	135	26.7 (24.25–28.73), higher in NAS 5	Higher insulin and HOMA-IR scores in NAS 5	ALL higher in NAS 5	Serum ferritin is higher in NAS 5 and serum CP lower	HDL lower in NAS 5
Suresh et al., 2024 [73]	7333	Higher percentage of obesity in normal ferritin MASLD	No difference	ALL higher in hyperferritinaemia	80% normal ferritin (<300/450 F/M mcg/L), 20% hyperferritinaemia	Higher percentage of hyperlipidaemia in the normal ferritin group with higher TC but higher TRG in hyperferritinaemia
Maliken et al., 2013 [88]	83	32.8 ± 6	26%	ALP is higher in patients with hepatocellular iron group	Ferritin is higher in the reticuloendothelial iron group, total ferritin population 225 (105–344)	No difference
Eder et al., 2020 [89]	299	28.7 (17.7–41.7), No difference amongst histological groups	NR	Highest ALT in the reticuloendothelial iron group	Highest serum ferritin in mixed pattern group	NR
Buzzeti et al., 2019 [90]	468	30.4 ± 5.8	29%, higher in NASH	ALT 65 (37–93), higher in NASH group	Ferritin 188 (61–314), 32% hyperferritinaemia, mixed deposition pattern higher in NASH	No difference between NASH vs. non-NASH
Adams et al., 2015 [93]	74	BMI 31.4, No difference	17.6%, no difference	ALT 48, AT 35, no difference	Ferritin 229, no difference	No difference
Yu et al., 2022 [97]	18,569	Higher in those who remained HCC-free	Higher rate in those who remained HCC-free	NR	No difference in serum iron, ferritin, or transferrin saturation	Higher rate in those who remained HCC-free
Yu et al., 2024 [69]	2340	33.37 ± 0.24	23.1%	NR	Serum ferritin: 180 ± 6, higher than non-SLD group, sTfR 27.45 ± 0.29%, lower in SLD group, no difference in serum iron	NR

### 3.2. From Steatosis to HCC: What Is the Role of MHF?

The landscape in steatosis-mediated HCC pathophysiology remains widely unexplored, with multiple pathways involved, especially in the MS population, with DM, activating multiple signalling pathways of cellular proliferation [98]. However, a few studies have outlined iron’s potential contribution to this process, especially under the prism of comparing it with other causes of chronic liver disease (CLD) and HCC-genesis. Various studies feature basic mechanisms of carcinogenesis and how iron overload is implicated. HepG2 cells overloaded with iron demonstrated epithelial-to-mesenchymal transition features, a key malignancy feature that awakens invasive potential. The E-cadherin differential expression profiles in this study might hint at its role in advanced HCC, showcased in previous clinical studies [99]. Another mechanism studied is DNA damage; in a study of diet-induced NASH in mice, mutY DNA glycosylase (MUTYH) knockout, a gene involved in oxidative stress DNA damage repair, was shown to increase oxidative stress markers and HCC-genesis, mediated by the Wnt/β-catenin pathway [100]. Studies of iron chelators have highlighted different biological checkpoints in HCC-genesis, where iron depletion can slow down cell proliferation [101] or metastatic potential [102], while the anti-tumour effects of deferasirox in cell lines and a murine model were not reproduced in a small case series of advanced HCC [103]. A clinical study of hepatic resection, stratified by cause of HCC, showcased that combining high serum ferritin levels with relative enhancement ratio in MR imaging can help accurately identify the NAFLD-HCC patients that have tumours with Wnt/β-catenin pathway activation, which may be resistant to immunotherapy due to the limited amount of lymphocytic infiltration [104].

The association of iron with HCC in SLD was clear in a large US cohort of long-term follow-up of NAFLD patients, where those with higher serum iron and transferrin saturation were at higher risk of disease progression and HCC development [97]. The same principle applies to NASH liver biopsies, as showcased in the classical study of Sorrentino et al., where semi-quantitative analysis of iron in liver biopsies showed higher scoring for HCC patients when compared to cirrhotic ones [105]. Interestingly, in 2017, Chung et al. showed that portal iron overload is associated with a worse prognosis after curative hepatectomy [106]. These two studies together showcase that iron overload in the porto-sinusoidal system can enhance HCC genesis, possibly via oxidative stress damage. Furthermore, studies of widely used clinical markers link high iron load with HCC development and progression. A preoperative serum ferritin concentration higher than 267 ng/mL was linked to a worse long-term prognosis (both overall and recurrence-free survival) in a Chinese cohort of HCC patients that underwent curative hepatectomy [107]. Most studies on the prognostic value of serum ferritin on HCC link hyperferritinaemia to adverse survival outcomes [108,109], except for a study by Uchino and colleagues that failed to connect elevated serum ferritin to worse prognosis in HCC patients post radiofrequency ablation [110]. The higher serum ferritin levels could reflect higher iron availability, a fundamental element for cellular division, whose presence is crucial for tumour growth [108,109,111]. The transferrin receptor has been identified as a key marker and mediator of iron-metabolism-related changes in HCC and was linked with worse prognosis after hepatectomy for HCC, probably secondary to the increased cellular invasion capacity of cells expressing it [112,113]. Its expression was higher in HCC cells compared to the surrounding normal liver tissue, and its effects could be either secondary to an mTOR-related pathway or by increasing resistance to ferroptosis [112,113]. Magnetic resonance imaging of HCC reveals that in cases of larger tumours, a peritumoral hypointense rim might be positively associated with the degree of fibrosis of the adjacent liver parenchyma and most probably reflects iron overload as a result of well-established CLD of several aetiologies, including SLD [114]. Finally, there is evidence that perioperative blood transfusion is independently associated with shorter survival and cancer recurrence after resection for HCC [115,116,117]. This is mainly due to the effect on immune function (T-lymphocytes and NK cells). However, iron overload could potentially have an effect.

Some interesting observations arise from patient cohorts with viral aetiology HCC in terms of iron metabolism profile and disease progression, demonstrating a significant diversity. There is ample evidence linking chronic HBV infection with excess serum ferritin and higher systemic iron availability, especially when HDV co-infection exists, while this is also reflected in the higher liver iron concentrations. Interestingly, iron excess shrinks with chronicity and disease progression and low serum iron is linked to worse survival in HBV-associated HCC. While iron could drive inflammatory processes in the initial phases of disease and mediate apoptosis, progression to cirrhosis and HCC are iron-demanding and can deplete iron storages [118,119]. This might be a common pattern as low serum iron is linked to worse prognosis in two different cohorts of HCC patients, one of HBV-related HCC and another of curative resection for mixed-cause HCC [119,120]. In the initial phases of HCV infection, increased hepcidin upregulates TfR1, an internalisation factor for HCV, while chronic disease is associated with lower hepcidin, higher systemic and lower cellular iron, which are not linked to disease progression, as opposed to the degree of steatosis, which is linked to fibrosis progression [121,122,123]. The abovementioned studies showcase differences in the role of iron in SLD versus non-SLD-related HCC; in the former, iron overload is often a driver of progression, viral infections might be associated with altering iron profiles as the disease initiates and then progresses. The key message remains that iron dysregulation remains an element of tissue damage and disease progression, albeit to an unknown extent [24,118]. Regarding haematological disorders, chronic secondary iron overload could also have an effect on the development of chronic liver disease and on hepatocarcinogenesis, as in the cases of patients with hereditary haemochromatosis or iron overload secondary to transfusions, but this has not been proven to follow the same pathophysiology as MASLD and MHF [31].

Systemic therapies for HCC are constantly evolving and while sorafenib, a multi-kinase inhibitor, has historically been the first targeted therapy for unresectable HCC [124], novel monoclonal antibodies have been introduced by different society guidelines as first-line treatments for the management of advanced HCC not amenable to locoregional therapies [125,126,127]. Immune cells have ligand–receptor immune checkpoints that can induce or attenuate immune response and immune checkpoint and undo the immune tumour evasion [128]. One of the most well-adopted therapies is the combination of atezolizumab, a programmed death ligand-1 (PD-L1) inhibitor, with bevacizumab, an anti-VEGF (vascular endothelial growth factor) monoclonal antibody [125,126,127,128,129], while other ICIs such as the combination of tremelimumab–durvalumab are quickly gaining their place in international guidelines.

Ferroptosis might come as a key player in increasing systemic therapy efficacy in MASLD-related HCC. Interestingly, while cancer cells are rich in iron and their survival largely depends on it, this could potentially induce ferroptosis, especially in the context of high lipid peroxidation rates, as in MASLD [130,131,132]. Thus, HCC cells utilise several mechanisms (mostly known antioxidants) to establish defences against ferroptosis [130]. While ferroptosis induction might lead to hepatocellular destruction, it could also impact the tumour microenvironment and its role remains ambiguous, especially in the non-tumorous steatotic liver [131,133]. Sorafenib initially seems to facilitate ferroptosis induction by an array of direct and indirect mechanisms, while continuous exposure might lead specifically mutated tumour cells to develop ferroptosis resistance [134]. While there might be data pointing to a poorer response of MASH-related HCC to ICI, hinting at the beneficial impact of ferroptosis induction [131], current EASL guidelines advise against considering the cause of liver disease in administering ICI for advanced HCC [125]. The body of evidence currently points to CD8+ T-cells and NK-cells as the main types of immune cells mediating an inflammatory response against HCC and induction of ferroptosis could be one of the mechanisms utilised [128,135]. Bearing this in mind and with a slowly growing body of evidence showing benefit in the use of ICI as adjuvant or even neo-adjuvant therapy, as well as more studies coming on the field, the role of ferroptosis on this process will certainly attract more research interest [128,129].

To some extent, further insight into this topic can be gained by combining observations from other iron and ferroptosis genetic studies in CLD and HCC patients that showcase different panels of ferroptosis-related genes and their impact on HCC survival [132,136,137,138,139]. Interestingly, in one of these panels [139], HRAS is included, a known promoter of SLD-HCC via immune pathway modulation. Different ferroptosis genes involved in HCC prognostic panels are shown in Table 2. It is important to highlight that not only ferroptosis-related genes have been linked to HCC. Natarajan et al. showed that H36D mutation of the HFE gene is associated with HCC development, mostly in metabolic-related pathways [140]. The percentage of HFE mutation HCC patients with further iron overload gene mutations reached 7.7% in a French cohort, reinforcing the role of genetic testing and hinting at potential therapeutic applications [141].

## 4. Conclusions

In this paper, we explored the impact of different degrees of iron metabolism dysregulation on the spectrum of steatotic liver disease and how they can ultimately lead to HCC development. The evidence shows that iron helps establish and propagate steatosis, as demonstrated in multiple cell and animal models and clinical studies, with involvement in different biological processes. Intense research currently focuses on the role of ferroptosis in this process of steatosis and its related HCC-genesis and has widened our understanding of this pathophysiological sequence. However, several questions remain unanswered and there are points worth exploring in the relationship between iron and liver steatosis, considering the evolving complexity of the hepatic microenvironment, which could insinuate the involvement of other biological processes where iron might be indirectly implicated. Overall, the above-presented knowledge helps assert the significant role of iron in SLD and HCC and can help integrate iron studies in everyday clinical practice, using clinical metrics of iron load, especially ferritin, in several stages of the disease spectrum to non-invasively monitor disease evolution as well as a trigger for clinical studies of different potential agents that could serve as adjuncts to already established therapeutic interventions for SLD or HCC.

## Figures and Tables

**Table 2 cancers-17-00842-t002:** Genetic panels of iron metabolism and ferroptosis-related genes in HCC patient prognoses.

Author, Year	Population Size	Genes Included	Biological Significance	Prognosis
Zhu et al., 2024 [137]	369	CYP3A5, SLC7A11, FLVCR1, PPAT, CYP2C9, IGSF3, G6PD, TFRC	Iron cellular influx and efflux, as well as utilisation	Worse overall survival for panel-positive patients
Shen et al., 2018 [142]	423	TFRC, FLVCR1, FTL	Iron cellular influx and efflux, as well as utilisation	TFRC, FLVCR1, FTL—overall survival, FTL—recurrence
Long et al., 2022 [143]	375	SLC7A11, SLC1A5, GCLM, SAT1	Ferroptosis inducers	High-risk group showed worse overall survival
Wang et al., 2022 [139]	631	SLC2A1, NRAS, HRAS, MAPK3, RRM2	Ferroptosis	High-risk group showed worse overall survival
He et al., 2023 [132]	310	G6PD, HRAS, NRAS, TIMM9, MYCN	Ferroptosis and immune	Survival and immunotherapy response
Wu et al., 2023 [138]	1077	CAPG, SLC7A11, S2STM1	Ferroptosis inducers and silencers	High-risk group showed worse overall survival and differences in immune check genes
Tang, 2020 [136]	370	ABCB6, FLVCR1, SLC48A1, SLC7A11	Ferroptosis	High-risk group showed worse overall survival, more somatic mutations, and altered immune tumour microenvironments

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
