# Peer review of "Exploring the Role of Metabolic Hyperferritinaemia (MHF) in Steatotic Liver Disease (SLD) and Hepatocellular Carcinoma (HCC)"

_cancers, 2025, doi:10.3390/cancers17050842_

Round 1

Reviewer 1 Report

Comments and Suggestions for Authors

The review is well written but i think a comment on the prognostic role of ferritin should be added. There are several reports in this regard, for example in patients undergoing TACE or RFA.

Some figures would improve the quality of the paper

THe authors should comment on the other potential confounders in this regard, for example drugs or, above all, diabetes mellitus (in this regard cite the comprehensive review PMID: 23845075 )

The authors should cite the most recent meta-analyses assessing the incidence of HCC in SLD.

Author Response

Dear Reviewer,

Thank you very much for your feedback and your useful comments. We have actioned them accordingly in the manuscript. In detail:

Comment 1: The review is well written but I think a comment on the prognostic role of ferritin should be added. There are several reports in this regard, for example in patients undergoing TACE or RFA.

Response: We have included a recent metanalysis on the utility of pre-treatment iron metabolism in HCC prognosis after treatment:Most studies on the prognostic value of serum ferritin on HCC link hyperferritinaemia to adverse survival outcomes [109,110] except for a study by Uchino and colleagues that failed to connect elevated serum ferritin to worse prognosis in HCC patients post radiofrequency ablation [111]. The higher serum ferritin levels could reflect higher iron availability, a fundamental element for cellular division, whose presence is crucial for tumour growth [109,110,112].”

Comment 2: Some figures would improve the quality of the paper.

Response: Due to the short time we had to prepare the revised version of the manuscript, we will not be able to produce any extra figures. However, we have made a graphical abstract which summarises the basic mechanisms the paper addresses, which we attach.

Comment 3: The authors should comment on the other potential confounders in this regard, for example drugs or, above all, diabetes mellitus (in this regard cite the comprehensive review PMID: 23845075)

Response: We have added a sentence highlighting the role of other factors ie DM in the pathogenesis of HCC: “with multiple pathways involved, especially in the MS population, with DM activating multiple signalling pathways of cellular proliferation [99]”

Comment 4: The authors should cite the most recent meta-analyses assessing the incidence of HCC in SLD.

Response: We have added text and references regarding the incidence of HCC in SLD: “with a US incidence of HCC in SLD of 0.72 per 1000 person-years[5], and Asian cohorts estimating that between 0.18 to 0.64 person-years[6,7]”

Looking forward to any extra feedback you might have.

Kind regards,

On behalf of the authors.

NA Anastasopoulos

Reviewer 2 Report

Comments and Suggestions for Authors

Authors aimed to explore  the  role  of  metabolic  hyperferritinaemia  (MHF)  in Steatotic  Liver  Disease  (SLD)  and  Hepatocellular  Carcinoma.    

(HCC). This is an interesting paper. There are several comments to be addressed.

1) Data regarding association between the needs of transfusion and adverse liver outcome, especially in patients with hematologic disease, should be addressed.

2) Is there any evidence about the benefit of iron-chelating agent ?

3) Ferritin might be suggested as one of biomarker for MASLD. It should be addressed further.

Author Response

Dear Reviewer,

Thank you very much for your feedback and your useful comments.  We have actioned them accordingly in the manuscript. In detail:

Comment 1: Data regarding association between the needs of transfusion and adverse liver outcome, especially in patients with hematologic disease, should be addressed.

Response: “There is evidence that perioperative blood transfusion is independently associated with shorter survival and cancer recurrence after resection for HCC [116-118]. This is mainly due to the effect on immune function (T-lymphocytes and NK cells). However, iron overload could also have an effect.”

“Regarding hematological disorders chronic secondary iron overload could also have an effect in the development of chronic liver disease and on hepatocarcinogenesis, as in the cases of patients with hereditary haemochromatosis or iron overload secondary to transfusions, but this has not been proven to follow the same pathophysiology as MASLD and MHF [31].”

Comment 2: Is there any evidence about the benefit of iron-chelating agent?

Response: We addressed the impact of iron chelating agents on SLD and HCC development by adding: “Iron chelators seem to demonstrate a consistent effect in attenuating fatty liver disease in multiple animal model studies. Dietary iron overload combined with high fat diet (HFD) induced more severe hepatic damage than HFD only, which was alleviated by deferoxamine [53], which has also been shown to have similar anti-inflammatory impacts when administered intraperitoneally [54]. Another study investigated the effect of deferoxamine on steatotic liver in both mice and cell lines and showed that extracellular iron reduction reversed ferroptosis features and reduced hepatocyte lipid deposition [55]. The combined action of deferoxamine and deferiprone shows a significant reduction in oxidative stress-related ischaemia reperfusion injury in rabbits hinting potential benefit in future research for steatotic liver disease reversal [56].” And “Studies of iron-chelators have highlighted different biological checkpoints in HCC-genesis where iron depletion can slow down cell proliferation [102] or metastatic potential [103], while the anti-tumour effects of deferasirox in cell lines and a murine model were not reproduced in a small case series of advanced HCC [104].”

Comment 3: Ferritin might be suggested as one of biomarkers for MASLD. It should be addressed further.

Response: Throughout the manuscript, we have sought to demonstrate how iron (and by extension serum ferritin) can be used a biomarker for disease severity in MASLD/MASH as well as a prognostic marker in post-treatment HCC survival, recapitulating and emphasising that in our final sentence (that I have highlighted, and I am quoting here for your convenience). For clarity, we have minimally modified our last period: “Overall, the above knowledge helps assert the significant role of iron in SLD and HCC and can help integrate iron studies in everyday clinical practice, using clinical metrics of iron load, especially ferritin, in several stages of the disease spectrum to non-invasively monitor disease evolution as well as a trigger for clinical studies of different potential agents that could serve as adjuncts to already established therapeutic interventions for SLD or HCC.”

Looking forward to any extra feedback you might have.

Kind regards,

On behalf of the authors.

NA Anastasopoulos

Round 2

Reviewer 1 Report

Comments and Suggestions for Authors

The manuscript is OK now. Thank you!

Reviewer 2 Report

Comments and Suggestions for Authors

Authors addressed raised issues appropriately.